# Arc-Mediated Synaptic Plasticity Regulates Cognitive Function in a Migraine Mouse Model

**DOI:** 10.3390/brainsci13020331

**Published:** 2023-02-15

**Authors:** Luyu Hu, Qiaoyu Gong, Yanjie Zhou, Yajuan Wang, Tao Qiu, Yuting Fang, Wanbin Huang, Jingjing Liang, Zheman Xiao

**Affiliations:** Department of Neurology, Renmin Hospital of Wuhan University, Wuhan University, No. 99, Zhangzhidong Road, Wuhan 430060, China

**Keywords:** migraine, cognitive impairment, activity-regulated cytoskeleton-associated protein (Arc), synaptic plasticity

## Abstract

Previous clinical and basic studies have shown that migraine is associated with cognitive impairment, anxiety, and depression. It severely affects the quality of life. In this study, C57BL/6 mice were randomly divided into four groups: IS group, IS+M group, and IS+S group with repeated application of dural inflammatory soup (IS) stimulation to establish a migraine model, followed by PBS, memantine, and sumatriptan interventions, respectively; the blank control group underwent the same treatment procedure but with PBS instead of IS and intervention drugs. The cognitive function of the mice was used as the main outcome indicator. After application of the IS, mice showed reduced pain threshold for mechanical stimulation, decreased learning memory capacity, attention deficit, a reduced number of dendritic spines in hippocampal neurons, and altered synaptic ultrastructure. The cognitive function indexes of mice in the IS+M group recovered with changes in Arc protein expression to a level not statistically different from that of the Control group, while the IS and IS+S groups remained at lower levels. The present results suggest that Arc-mediated synaptic plasticity may be an essential mechanism of cognitive dysfunction in migraine.

## 1. Introduction

The current mainstream treatment plan for migraine focuses on pain but seems to ignore the other symptoms and deep effects of migraine. Approximately 2.5% of people with migraine progress to a condition where headaches occur 15 days or more per month, of which eight or more days of headaches and accompanying symptoms conform the diagnostic criteria for migraine, which we call chronic migraine (CM) [1,2]. Due to its many symptoms, such as frequent headaches, nausea, vomiting, cognitive impairment, anxiety, and depression, CM places a huge burden on people. Among them, cognitive deficits play a significant role in the individual experience of migraine and serve as a factor in attack-related disability, which can affect daily functioning at work, home, and in social situations as well as managing their attacks. Cognitive symptoms are second only to pain in terms of intensity and attack-related disability [3,4]. Therefore, it would be a good direction to take to improve cognitive function into account, rather than just focusing on pain, when developing new drugs for migraine.

We still know very little about how migraine causes cognitive impairment. Since synaptic plasticity is crucial for cognition [5], migraine-induced cognitive impairment may be closely related to abnormal synaptic plasticity in different brain regions. For example, researchers found increased synaptic plasticity in the trigeminal nucleus caudalis (TNC) of inflammatory soup (IS) model rats, leading to central sensitization and nociceptive hyperalgesia [6,7]. Another research team found reduced synaptic plasticity in IS rats’ hippocampus; accordingly, these rats showed spatial memory deficits in the Morris water maze [8]. How the pathophysiological processes of migraine cause altered synaptic plasticity and why different brain regions are affected in opposite ways are questions that deserve further exploration.

Synaptic plasticity includes Hebbian plasticity (such as LTP and LTD) that responds rapidly within minutes and homeostatic synaptic plasticity (HSP, including synaptic scaling, presynaptic modulation, and postsynaptic modulation) that can continue to function. Because Hebbian plasticity has a positive feedback mechanism and a tendency to self-reinforce, its modification on glutamatergic synaptic strength usually needs to be maintained or limited by HSP, which depends on RNA transcription, protein synthesis and degradation to play its role [9,10,11,12]. Arc (activity-regulated cytoskeleton-associated) protein, an immediate early gene product, is thought to be a hub for the transition from rapid response to long-term regulation of synaptic plasticity. Multiple lines of evidence points to Arc’s essential role in stabilizing LTP, LTD, and the establishment of synaptic homeostasis [13,14]. Furthermore, Arc has gained widespread attention in several neurological disorders, such as dementia, psychiatric and neurodegenerative diseases [9,10]. Therefore, probing the alterations and role of Arc in migraine seems to be a promising direction to explore the mechanisms of changes in synaptic plasticity in migraine.

Based on the above theory, in this study, a randomized controlled trial was designed in which animals were randomly placed into four groups. The IS, the IS+M, and the IS+S groups, where a migraine model was established by repeated application of dural IS stimulation, were followed by PBS, memantine, and sumatriptan interventions, respectively; the blank Control group underwent the same treatment procedure but with PBS instead of IS and intervention drugs. The cognitive function of the animals was used as the primary outcome indicator. We hypothesized that migraine affects cognitive function through Arc-mediated synaptic plasticity, and then, this alteration should be somewhat restored by modulation of Arc expression. In addition, we want to test whether current first-line drugs for migraine can have an effect on this cognitive impairment.

## 2. Materials and Methods

### 2.1. Animal

This study used adult male C57/BL mice (22–28 g). Male mice were chosen because estrogen fluctuation influences the occurrence and frequency of migraine. The Experimental Animal Center of the Renmin Hospital at Wuhan University provided all of the animals, and they were all kept in the specific pathogen-free (SPF) environment. Prior to the procedure, the mice were kept in a 12/12 h light-dark cycle at 23 °C ± 1 °C for a week, with free eating and drinking. Researchers gently handled the mice for 5 min per day. All experiments adhered to the recommendations of the International Association for the Study of Pain in Conscious Animals. We conduct experiments with the fewest possible animals and minimize the severity of their suffering.

### 2.2. Grouping and Modeling

No prior statistical calculations of sample size were performed. We designed to divide the mice into four groups, each with four experimental projects to perform. In order to meet the requirement of at least three individual mice per group for each project (at least six for the Western blot project), we needed 60 mice, with an additional eight mice added to prevent losses due to adverse events, and one mouse excluded during rearing due to substandard weight, for a final total of 67 mice participating in the experiment.

All mice were numbered and randomly divided into four groups using the random number function of Microsoft Excel as follows: the Control (n = 17), the IS (n = 17), the IS+M (n = 17), and the IS+S (n = 16).

After induction of anesthesia, all mice were inhaled with isoflurane (3% concentration, 300 mL/min flow) to maintain anesthesia, placed in a stereotaxic apparatus, and disinfected with the iodophor. Along the mouse’s cerebral midline, a cut is made. Then, we used a cranial drill to drill a bone window with a diameter of 1 mm to expose the dura mater and be careful not to damage it. According to the stereotactic coordinates, use dental cement to fix a sleeve on the bone window and suture the wound. After surgery, the mice recuperate for a week. All mice received prophylactic treatment with antibiotics after the operation. Then, 20 μL of Inflammatory Soup (IS) was administered to mice in the IS, IS+M and IS+S groups through the cannula every day for four days. The Control group, in turn, received 20 μL of phosphate-buffered saline (PBS) injections. We ensured that the cannula and bone window were not blocked so the drug can directly contact the dura mater. Half an hour before each IS infusion, mice in the IS+M group were given an intraperitoneal injection of memantine (1 mg/mL concentration, 10 mg/kg) [15,16]; after the IS infusion, the IS+S group was given a subcutaneous injection of sumatriptan (1.2 mg/mL concentration, 0.6 mg/kg); the IS and the Control were assigned PBS. The IS used in this study was made by dissolving 1 mmol/L bradykinin, 1 mmol/L serotonin, 1 mmol/L histamine, and 0.1 mmol/L prostaglandin E2 in PBS (pH 7.4).

### 2.3. Behavioral Assessment

Behavioral trainings were performed after daily administration to acclimate the mice to the test site and apparatus. All behavioral tests were performed after four days of continuous dosing. Then, the mice were euthanized, and brain tissue was extracted for subsequent analysis. All animal behavior assessments were performed blindly between 10 a.m. and 4 p.m.

### 2.4. Mechanical Pain Threshold

For the mechanical pain threshold assessment, mice were placed in a 30 cm × 20 cm × 20 cm cage allowing free movement, and the von-Frey filaments of different strengths (0.002, 0.008, 0.07, 0.16, 0.4, 0.6, 1.0, 1.4, 2.0 g) were applied vertically to their periorbital region (Figure 1A). After the stimulation, the mice’s head retracted or vocalized to the other side, raised the ipsilateral front paws above the eyes, and touched its face, indicating a positive response. Facial pain from mechanical stimulation was assessed, and a response threshold was calculated.

### 2.5. Facial Wiping

The number of facial wipes was counted for 1 h immediately after the last IS infusion. Only unilateral forelimb wiping behaviors were calculated, excluding grooming and scratching behaviors. These forelimb wipes are usually gentle single strokes starting at the back of the cheek and moving forward in a caudal to cephalad direction. There was no stroking back and forth, which suggests scratching. Compared to the more vigorous scratching of the hind limbs, these scratches are brief and may take less than 0.5 s to complete. Contact during wiping appears to be related to the medial side of the forearm. The paws do not touch the cheeks and face forward rather than face towards the cheeks (Figure 1B). In contrast, grooming behavior consisted of wiping both forelimbs simultaneously, alternating rubbing, and wiping the cheeks starting behind the ears [17].

### 2.6. Novel Object Recognition (NOR)

There are two phases in this test (Figure 2A). Stage 1 (T1): Place two identical objects (objects 1 and 1’) in the center of the two opposite areas at an opening 40 cm × 20 cm × 20 cm cage, and place the mice alone in the cage for 10 min. After the exploration, the mice were taken out and put back into the breeding cage. Clean the observation box to eliminate the smell. Stage 2 (T2): After a certain period, the mice were placed in the opening field to explore for five minutes while object 1 was left unchanged (old object) and object 1’ was switched out for object 2 (novel object). After the experiment, the observation box was thoroughly cleaned up. Use the animal tracking system to record and analyze the time of the mice to explore two objects, respectively, in stages T1 and T2. Object 1’s exploration time was noted as F, object 1’s or object 2’s exploration time was noted as N, and the combined exploration time of the two items was noted as F + N. The distinguish ratio (DR, DR = N ÷ (N + F) × 100%) and the distinguish index (DI, DI = (N − F) ÷ (N + F) × 100%) were calculated. Exploration is defined as a mouse facing an object with its nose less than or equal to 2 cm apart from the object.

### 2.7. Attentional Set-Shifting (ASS)

Next, we trained mice to dig into the bowl for food. The bait was deposited in one of two bowls, varying the material, odor or medium of the bowl, and the mice were subjected to three direct discrimination tests. Each mouse was trained to make six correct choices in a row for each criterion, with the mouse digging into the first bowl being defined as a single choice. Mice were kept hungry before the formal test to keep them motivated to dig.

The ASS was performed after the completion of the drug intervention procedure. Mice were tested in seven stages: simple discrimination (SD), compound discrimination (CD), reversal of CD (R1), intradimensional shift (IDS), second reversal (R2), extradimensional shift (EDS) and reversal of EDS (R3). One factor of the bowl is changed at each stage to test the mice’s capacity to filter out disturbances while searching for food. Each mouse underwent the same test procedure. The mice’s trails to criterion (TTC) at each test stage were recorded for data analysis.

### 2.8. Cell Culture

SH-SY5Y cells were purchased from the China Center for Type Culture Collection (CCTCC, Wuhan University, Wuhan, China) and maintained in a complete medium. The complete medium was made from the DMEM high-glucose medium supplemented (Cytiva, Shanghai, China) with 10% fetal bovine serum (Cytiva, China), 100 units/mL penicillin, and 100 mg/mL streptomycin. They were maintained in a humidified 5% CO_2_/95% air environment at 37 °C. The medium was changed every other day, and cells were passaged every three days. The number of cell passages used in all experiments is less than 17.

SH-SY5Y cells were placed into three groups (the NGF, the N + M, and the control) and inoculated into 6-well plates with 1 million cells/well. NGF 50 ng/mL was added to the complete medium of the NGF group; meanwhile, NGF 50 ng/mL and memantine 50 μM were added to the complete medium of the N + M group. Then, 6-well plates were placed in the incubator and continued to incubate for 24 h.

### 2.9. Western Blot (WB)

After model establishment and drug intervention were completed, fresh cerebral tissue was isolated from mice, and cells were collected from the culture medium. Both of them were processed with the same procedure to extract the protein. First, we immersed hippocampal and cortical tissue or cells in protein extract (97% RIPA lysis buffer, 2% protease inhibitor cocktail, 1% PMSF); then, we incubated on ice for 15 min, sonicated for 5 min, and placed the homogenate on ice. After 30 min, we centrifuged the protein homogenate and collected the supernatant.

Proteins extracted from animals or cells were subjected to the same procedure for gel electrophoresis. The proteins were separated on SDS-PAGE gels and then transferred to PVDF membranes. The membranes were closed with skimmed milk and then incubated with the primary antibody overnight. In this study, we detected Arc (1:1000; Santa Cruz, Dallas, TX, USA), NR2B (1:1000; Santa Cruz, USA), NR1 (1:1000; Fine Test), GluR-1 (1:1000; Santa Cruz, USA), GluR-2 (1:500; Santa Cruz, USA), synaptophysin (SYP, 1:1000; Wanleibio, Shenyang, China). Since the molecular weight of SYP is close to that of GAPDH, we use two control reference proteins, β-actin (1:3000; Servicebio, Wuhan, China) and GAPDH (1:3000; Servicebio, China). Protein levels were normalized. The next day, membranes were incubated with the horseradish peroxidase-conjugated anti-rabbit or anti-mice secondary antibody (1:3000; Servicebio, China). Finally, we used Omni ECL reagent (Epizyme Biomedical, Shanghai, China) to visualize the antibody-reactive bands. Images were acquired using a chemiluminescence imaging system (BIO-RAD ChemiDoc Touch, USA) and analyzed using Image Lab (version 6.1).

### 2.10. Immunofluorescence Staining

Mice were anesthetized and perfused with physiological saline through the heart, and 4% paraformaldehyde was perfused after emptying the blood. After perfusion, the telencephalon of the mice was isolated and fixed; then, it was dehydrated with alcohol (70–99.7%). The samples were washed by soaking in xylene for 1 h; then, the tissue was immersed in molten paraffin for 3 h in a 60 °C oven. The paraffin-immersed tissue was rapidly cooled in an embedding frame filled with molten paraffin, and the tissue was cut into 3 μm thick slices with a rotary microtome (Leica HistoCore BIOCUT, Germany) and fixed on an adhesive glass slide.

For immunofluorescence staining, sections were dewaxed using xylene and rehydrated using 95–70% alcohol and ddH_2_O. After hydration, the sections were subjected to antigen repair and permeabilization for 20 min at room temperature. Then, the sections were closed and incubated overnight with the following primary antibody: mice anti-mice Arc (1:100; Santa Cruz, USA), mice anti-mice GluR1 (1:100; Santa Cruz, USA), and rabbit anti-mice SYP (1:100, Wanleibio, China). The next day, sections were incubated at 37 °C for 90 min in the dark with species-specific fluorophore-labeled secondary antibodies: Dylight 488 conjugated to goat anti-mice immunoglobulin G (IgG, 1:200, Servicebio, China) antibody and Dylight 594 conjugated to goat anti-mice IgG (1:200, Servicebio, China) antibody. Cell nuclei were stained with 4′,6-diamidino-2-phenylindole (DAPI) for 10 min. Finally, sections were sealed with an anti-fluorescence quenching mounting medium (Servicebio, China). Then, the sections were observed and photographed with an upright fluorescence microscope (OLYMPUS BX43, Japan). Arc and SYP were observed in the hippocampus and prefrontal cortex with a 20× and 40× objectives lens. Images were analyzed using ImageJ (version 1.8.0_172). The relative intensities of Arc and SYP fluorescence were compared by mean optical density.

### 2.11. Golgi–Cox Staining

The Golgi–Cox staining was carried out using a method of Robbin et al. [18]. After euthanizing the mice, their brain was immediately isolated and fixed in a 4% paraformaldehyde solution. After 24 h, it was cut into 2–3 mm tissue blocks in a coronal position. We rinsed the tissue block several times with normal saline, placed it in a specimen bottle, added Golgi staining solution to the bottle and immersed the tissue block completely. Then, we put it in a ventilated place away from light. We changed the dye solution for the first time at 48 h and then changed the dye solution every 72 h after that; it was processed for 14 days. The tissue blocks were successively dehydrated in 15% and 30% sucrose solutions. Next, the tissue block was taken out, treated with concentrated ammonia water for 45 min, treated with acid film fixer for 45 min, and washed with ddH_2_O between steps. Then, it was dehydrated, cut into 100 μm sections using a cryostat (Leica CM1950, Germany), and fixed on glass slides. We used gradient concentration alcohol (70–99.7%) to dehydrate, mounted with glycerol gelatin, and then used upright optics microscope (NIKON Eclipse E100, Japan) multi-slice scanning to acquire pictures, which were analyzed by the ImageJ software mentioned above.

### 2.12. Transmission Electron Microscope (TEM)

Brain tissue was isolated intact within 1 min after the death of the mice and immediately immersed in pre-chilled 2.5% glutaraldehyde (Sevicebio, China) at 4 °C. Hippocampal tissue was isolated and evenly cut into many 1 cubic millimeter squares. Then, each group of 5 seahorse tissue squares was placed in specimen bottles containing 2.5% glutaraldehyde and fixed for 4 h. The fixed samples were sent to the Electron Microscopy Center (Renmin Hospital of Wuhan University, Wuhan, China) for the next step of fabrication and TEM scanning by professional staff. Images were acquired using a transmission electron microscope (Hitachi HT7800, Japan) and its imaging system, and using the ImageJ described above, we analyzed the images.

### 2.13. Statistical Analysis

In the study, all data are primary data and presented as means (±SEM). All statistical analyses were performed using GraphPad Prism Software (version 8.0.2). Two-way analysis of variance (ANOVA) with Tukey’s multiple comparisons test or Sidak’s multiple comparisons test were used to analyze differences between multiple groups at different times or stages. One-way ANOVA with Tukey’s multiple comparisons test was used to compare quantitative data between multiple groups. All values of *p* < 0.05 were considered statistically significant.

## 3. Results

Throughout the experiment, we lost three mice due to anesthesia and surgical accidents, and the total number of mice that survived and were successfully molded was 64:16 each for the Control, the IS, the IS+M, and the IS+S. All mice in each group participated in behavioral tests. Cortical and hippocampal tissues were taken from seven mice (7/16) per group for WB, brain sections were taken from three mice (3/16) per group (20 sections from a single mouse) for immunofluorescence, intact brains were taken from three mice (3/16) per group for Golgi–Cox staining, and five tissue blocks from the hippocampus of each of the three mice (3/16) per group were taken for TEM.

### 3.1. Mechanical Threshold and Spontaneous Head Scratching

Assuming that repeated dural IS stimulation can cause mice to develop migraine features, then IS mice should exhibit decreased mechanical pain thresholds and increased spontaneous pain behavior.

Mice were stimulated with von-Frey filaments to stimulate the measurement area (Figure 1A) to evoke their pain response, starting at low intensity, and the intensity of the mice’s first avoidance behavior was recorded as the pain threshold. Pain thresholds were significantly different among the four groups (*p* < 0.0001, two-way ANOVA, Figure 1). There was no significant difference among the four groups on the first day of drug delivery (*p* = 0.0728, one-way ANOVA). From the second day of administration, the pain threshold was significantly lower in the IS group (*p* = 0.0038, two-way ANOVA with Tukey’s test) and the IS+S group (*p* = 0.0108, two-way ANOVA with Tukey’s test) than in the Control group. On the third day, as the pain threshold decreased further, the IS and IS+S groups also began to show differences from the IS+M group (*p*-values of 0.0077 and 0.0021, respectively; two-way ANOVA with Tukey’s test). The difference reached its maximum on the fourth day of dosing, with the IS group being lower than the Control group (*p* < 0.0001, two-way ANOVA with Tukey’s test) and the IS+M group (*p* = 0.0010, two-way ANOVA with Tukey’s test), as did the IS+S group (*p* values of 0.0001 and 0.0032, two-way ANOVA with Tukey’s test). There was no significant difference between the IS+M and the Control (*p* = 0.7756, one-way ANOVA).

After completing the administration procedure for four consecutive days, the number of cheek wipes in the four groups showed significant differences (F = 76.53, *p* < 0.0001, one-way ANOVA, Figure 1). Respectively, the number of the four groups is 24.50 ± 5.89 (Control), 15.67 ± 5.21 (IS+M), 141.60 ± 11.61 (IS), and 18.00 ± 3.74 (IS+S). Among them, the number of the IS increased significantly compared with those of the Control (*p* < 0.0001, one-way ANOVA with Tukey’s test), the IS+M group (*p* < 0.0001, one-way ANOVA with Tukey’s test), and the IS+S group (*p* < 0.0001, one-way ANOVA with Tukey’s test). In comparison, both IS+M and IS+S groups were not significantly different from the Control (*p*-value 0.9999 and 0.9060, respectively, one-way ANOVA with Tukey’s test).

### 3.2. Novel Object Recognition Test and Attention Set-Shifting Test

As the assay test for animals’ cognitive function, NOR reflects the recognition memory function of animals, while ASS examines the ability of animals to form, maintain and shift attention and reverse learning function. If memantine and sumatriptan can restore cognitive dysfunction in IS mice, then the IS+M group and IS+S group’s DR and DI, and their TTC in IDS, REV, and EDS stages should be statistically significantly different from those of the IS group.

For NOR (Figure 2), at the T1 stage, the exploration time for objects 1 and 1’ show no difference in each group of experimental animals (*p* = 0.6479, two-way ANOVA). However, the sum of the time spent exploring objects 1 and 1’ differed between groups of mice. The time of exploring in the Control group, the IS+M and the IS+S was significantly longer than that of the IS (*p* = 0.0004, 0.0002, 0.0287, two-way ANOVA with Sidak’s test). At the T2 stage, the DR of the Control and the IS+M was significantly greater than that of the IS (*p* = 0.008, 0.0023, two-way ANOVA with Tukey’s test) and the IS+S (*p* < 0.0001, <0.0001, two-way ANOVA with Tukey’s test). It can be observed that the DI has the same trend (*p* = 0.0004, 0.0046, 0.0054, 0.0438, two-way ANOVA with Tukey’s test).

For the ASS (Figure 3), the three stages SD, CD, and IDS indicated the formation and maintenance of attentional fixation, and there were significant differences between the four groups in all three stages (7.26, 20.31; *p* = 0.0010, 0.0114, 0.0004; one-way ANOVA). During the IDS phase, the TTC of the IS was greater than that of the Control, IS+M, and IS+S groups (*p* < 0.0001, <0.0001, <0.0001, two-way ANOVA with Tukey’s test). The EDS phase reflected the ability to shift attention, where the TTC of the IS was still more remarkable than that of the Control and the IS+M (*p* < 0.0001, <0.0001, two-way ANOVA with Tukey’s test), and the IS+S group showed the same trend (*p* = 0.0030, = 0.0478, two-way ANOVA with Tukey’s test). Finally, during the REV phase, an increase in TTC indicated impaired reversal learning, with greater TTC in the IS and IS+S groups than in the Control and IS+M groups during the REV3 phase (*p* = 0.0030, =0.0011, =0.0081, =0.0030; two-way ANOVA with Tukey’s test).

### 3.3. Molecular Detection

Assuming that memantine regulates cognitive function by modulating the expression of Arc, the expression of Arc and its related proteins should correlate with the performance of the cognitive function in mice. That is, the overexpression of Arc can be detected in mice with cognitive dysfunction, and inhibition of this overexpression can restore cognitive function. In addition, if Arc is artificially upregulated, it can cause similar changes in cognition as in IS model mice.

In the brain regions most closely related to cognitive function, the hippocampus and prefrontal cortex, we used WB and immunofluorescence to probe the expression of Arc and related proteins.

WB’s results from hippocampal and cortical tissue show (Figure 4 and Figure 5) that the expression of NR2B in the hippocampus of the IS was significantly higher than that of the Control group and the IS+M group (*p* = 0.0239, 0.0320, one-way ANOVA with Tukey’s test). Similar to this, the Arc in the hippocampus of the IS group was higher than that of the Control group and the IS+M group (*p* = 0.0167, 0.0188, one-way ANOVA with Tukey’s test). In contrast, GluR1 expression was significantly higher in the Control group and the IS+M group than in the IS group (*p* = 0.0068, 0.0474, one-way ANOVA with Tukey’s test), while the expression of SYP was similar to it (*p* = 0.0166, 0.0023, one-way ANOVA with Tukey’s test). The results in the prefrontal cortex were consistent with the hippocampus. The levels of NR2B and Arc were suppressed in the IS+M group compared to the IS group (*p* < 0.0001, =0.0056, one-way ANOVA with Tukey’s test), while the expression of GluR1 was higher than in the IS group (*p* < 0.0001, one-way ANOVA with Tukey’s test). After the introduction of sumatriptan treatment (Figure 5), the IS+S group showed no significant difference in protein expression in the prefrontal cortex and hippocampus compared to the IS group (*p*-values for NR2B, Arc, and GluR1 in the prefrontal cortex compared to the IS group were 0.9888, 0.1202, and 0.2654, respectively; *p*-values for NR2B, Arc, and GluR1 in the hippocampus compared to the IS group were 0.7432, 0.5669, 0.8727; one-way ANOVA with Tukey’s test).

Immunofluorescence for animal prefrontal cortical and hippocampal tissue sections is shown in Figure 6, Figure 7, Figure 8 and Figure 9. Arc is stained with a green fluorescent dye. SYP and GluR1 are stained with a red fluorescent dye. Arc is distributed in both the nucleus and cytoplasm, so it appears as a green glowing mass around or overlapping with the nucleus. As membrane proteins, SYP is represented by the distribution of red light spots around the nucleus. In the hippocampal CA1 area and DG area, the expression of Arc in the IS group was significantly higher than that in the Control group and the IS+M group, as seen from the comparison of the mean gray values (*p* = 0.004, 0.006, one-way ANOVA with Tukey’s test). For SYP, the expression of CA1 in the IS group was significantly lower than that in the Control group and the IS+M group (*p* < 0.0001, =0.0062, one-way ANOVA with Tukey’s test). In the CA3 region, no significant differences were observed in Arc and SYP among the three groups (*p* = 0.0605, 0.2643, one-way ANOVA). GluR1 expression in the hippocampal CA1 region was also much more significant in the IS+M group than in the IS group (*p* < 0.0001, one-way ANOVA with Tukey’s test). After the introduction of sumatriptan treatment (Figure 8 and Figure 9), the IS+S group showed no significant difference in the hippocampal CAI region Arc compared to the IS group (*p* = 0.9448, one-way ANOVA with Tukey’s test), nor was GluR1 expression more than in the IS group (*p* = 0.8041, one- way ANOVA with Tukey’s test). The protein immunofluorescence in the prefrontal cortex was the same as in WB.

In the Western blot assay of cell cultures, we detected more proteins upstream and downstream of Arc (Figure 10). Similar to the animal protein analysis, the expressions of NR2B and Arc in the NGF group were significantly higher than those in the Control group and the N+M group (*p* < 0.0001, <0.001, one-way ANOVA with Tukey’s test), while the expressions of GluR1 were significantly lower than those in the Control and the N+M groups (*p* < 0.0001, <0.0001, one-way ANOVA with Tukey’s test). The expression of SYP in the NGF group was similarly lower than that in the Control and N+M groups (*p* < 0.0001, =0.0002, one-way ANOVA with Tukey’s test). However, NR1 and GluR2 were not significantly different among the three groups (*p* = 0.9691, 0.8961, one-way ANOVA).

### 3.4. Synapse Morphology and Structure

Synapses are critical to transmitting information by neuronal cells and are the basis of cognitive function. Cognitive dysfunction is often associated with synaptic damage, and restoring damaged prominence is essential to restoring cognitive function. Assuming that memantine or sumatriptan can restore cognitive function in IS mice, the corresponding synaptic damage should also be repaired.

We observed dendrites in the hippocampus of mice (Figure 11). In the hippocampal CA1 region, the density of dendritic spines per 20 μm in the IS (7.0 ± 2.1) was significantly lower (*p* = 0.0015, 0.0293, one-way ANOVA with Tukey’s test) compared with the Control (14.3 ± 0.67) and the IS+M (10.67 ± 0.88). However, the differences among the three groups in the hippocampal CA3 and DG regions were not statistically significant (*p* = 0.1054, 0.5865, one-way ANOVA). Additionally, there was no discernible difference between the IS+S receiving sumatriptan treatment and the IS in the number of dendritic spines in the hippocampal CA1 region (*p* = 0.9948, one-way ANOVA with Tukey’s test).

The ultrastructure of mouse hippocampal synapses was observed using TEM (Figure 12). We found that the presynaptic membrane, postsynaptic membrane, and synaptic vesicles (SV) were visible in the Control group and IS+M group, while the postsynaptic membrane in the IS group became blurred and synaptic vesicles were more challenging to identify. Observations of synaptic ultrastructure mainly assessed changes in synaptic cleft width, active zone length, synaptic vesicle number, and postsynaptic density (PSD) thickness. Compared with the Control and the IS+M groups, PSD thickness (*p* = 0.0008, 0.0015, one-way ANOVA with Tukey’s test) and synaptic vesicle numbers (*p* = 0.0460, 0.0025, one-way ANOVA with Tukey’s test) were reduced in IS group. However, there was no significant difference in the length of the synaptic active area and the width of the synaptic cleft among the three groups (*p* = 0.0924, 0.1338, one-way ANOVA). In addition, there was no significant difference in hippocampal synaptic PSD thickness and SV number in the IS+S group treated with sumatriptan compared with the IS group (*p* = 0.9998, 0.9994, one-way ANOVA with Tukey’s test).

## 4. Discussion

### 4.1. Repeated Administration of Dural Inflammatory Stimuli Induces Cognitive Impairment in Mice

In the IS group of mice, we observed a sustained decrease in mechanical pain thresholds and an increase in facial wiping behavior, the latter being considered a marker of spontaneous pain behavior [17]. Based on these, a migraine model was successfully established. In a new object recognition experiment, migraine model mice showed significantly lower recognition ratios and recognition indices for both new and old objects than controls, suggesting that migraine causes a decrease in recognition and memory abilities [19]. At the same time, the attention shift test indicated that the power of migraine rats to form, maintain and divert attention and even reverse learning was impaired. These are consistent with the clinical observations that have been reported [20,21]. They also appear to be less exploratory (reduced exploration time), and, not coincidentally, a study conducted in a rhesus monkey migraine model found that migraine led to reduced motor behavior, reduced time for daily activities, and reduced feeding [22]. The above evidence seems to prove that migraine does cause disabling effects in patients, both in animal models and in clinical patients.

### 4.2. Hippocampal Synaptic Plasticity Is Impaired in Cognitively Impaired Mice

At the molecular level, we found, unsurprisingly, increased Arc expression in the cortex and hippocampus in migraine mice, especially in the hippocampal CA1 and DG regions, where there are numerous pyramidal cells and synaptic connections. It has been found that the expression of Arc is mainly regulated by two forms: N-methyl-D-aspartate receptor (NMDAR) can activate the cyclic adenosine monophosphate/protein kinase A (cAMP/PKA) pathway leading to the upregulation of Arc protein, and neurotrophic factors through mitogen-activated protein kinase (MAPK) pathway induce increased Arc mRNA transcription and protein expression [23,24,25]. The existence of NMDAR in migraine has long been a broad concern [26], and the research teams mentioned above have found evidence of NR2B upregulation and enhanced synaptic plasticity in trigeminal spinal tract nuclei of migraine models [6,7]. Although there is no evidence of elevated NR2B in the hippocampus, reduced synaptic plasticity has been found in the hippocampus [8]. NR2B in the hippocampus was significantly overexpressed in our migraine mice. It is undeniable that NMDAR itself induces pain responses and changes in synaptic plasticity [27,28,29]. However, it is difficult to explain that the same increase in NR2B in different brain regions causes opposite changes in synaptic plasticity. Therefore, Arc may be the critical factor.

The Arc protein exists specifically in neurons of the dorsal hippocampus, primary somatosensory cortex, and dorsal striatum [30]. Arc proteins regulate synaptic plasticity in several forms. Arc can bind to endophilin-3 (Endo3) and dynamin 2 (Dnm2) and participate in clathrin-mediated α-amino-3-hydroxy-5-methyl-4-isoxazolepropionic acid receptor (AMPAR) endocytosis [31,32]. It has been found that the overexpression of Arc in different phenotypes may lead to increased or decreased dendritic spines, which seems to correlate with its ability to interact with endophilin 3 [33]. Arc can also regulate AMPAR transport and localization in the PSD by interacting with clathrin adapter protein 2 (AP-2) and transmembrane AMPAR regulatory protein TARPγ2 (stargazin) [34,35]. In the nucleus, Arc binds to the β-spectrin isoform βSpIVΣ5 and associates with promyelocytic leukemia (PML) nuclear bodies in the nuclear matrix relying on interaction with actin-binding proteins; this interaction promotes the downregulation of GluR1 transcription and AMPAR entire dendrite steady-state scaling [36,37]. AMPARs are direct effector molecules of changes in synaptic function and plasticity [38]. Arc by itself can induce membrane curvature and budding [39]; this may be how it participates in the vesicle pool cycle to regulate the number of presynaptic vesicles. Arc is involved in processes such as the structural plasticity of dendritic spines and LTP consolidation, but these functions are not well understood and require further study [40,41]. When Arc is overexpressed, the presynaptic function, postsynaptic membrane function, morphology, structure, and number of synapses will be affected by the above effects [9,42,43]. In the hippocampus of migraine mice, we detected a significant downregulation of both APMAR (especially GluR1) and SYP, implying a downregulation of presynaptic function and synaptic scaling at this site. We demonstrated a reduction in the number of dendritic spines, a reduction in presynaptic vesicles, and a reduction in the thickness of the postsynaptic membrane PSD in neuronal cells in the CA1 region of the hippocampus in Golgi staining and TEM images. Since the expression of Arc and related proteins in the prefrontal cortex shows a similar trend to that of the hippocampus, the same impairment of synaptic plasticity in the prefrontal cortex may exist.

### 4.3. Induction of Arc Overexpression in Neuronal Cells Causes Molecular Changes Similar to Those in Cognitively Impaired Mice

We performed cellular and drug intervention experiments to verify whether the active regulation of Arc expression can influence the pathophysiological process of synaptic plasticity changes in migraine. Nerve growth factor (NGF) was selected to upregulate Arc expression, while SH-SY5Y cells, which do not express TrkA receptors and are not affected by other effects of NGF, were chosen as a model [44,45]. For the choice of intervention drugs, we used memantine, a non-competitive inhibitor of NMDAR, to inhibit Arc upregulation, as NMDAR inhibitors can block any form of Arc overexpression [15,25]. In NGF-treated neuronal cells, we saw an upregulation of Arc and AMPAR expression and a significantly higher level of NR2B expression than in the control group. Furthermore, NGF-induced Arc overexpression was prevented after intervention with memantine, while NR2B overexpression was restored. This seems to suggest a deeper mechanism for the NMDAR-Arc-AMPAR triple interaction, and in the future, exploring this regulatory mechanism may lead to surprising findings. However, this cellular experiment is also limited in that we have only verified the relationship between upstream and downstream regulation of Arc in neurons. Evidence that synaptic plasticity can be directly altered by modulating this series of molecules is still lacking. In possible further experiments, it would be an excellent direction to use primary hippocampal neuronal cells as a model to investigate the effect of modulating Arc on synaptic plasticity.

### 4.4. Arc Inhibition with Memantine Restores Cognitive Impairment and Synaptic Plasticity to Varying Degrees in Mice, but Sumatriptan Has Little Effect

In migraine mice, we gave memantine by intraperitoneal injection, and this drug could cross the blood–brain barrier. After Arc and NR2B overexpression in the hippocampus of migraine mice was blocked, the expression of SYP and AMPAR was restored; the number of dendritic spines in hippocampal neurons, the number of presynaptic membrane vesicles, and the thickness of postsynaptic membrane PSD were restored to levels comparable to those of the control group; and the animals’ recognition and memory abilities were meaningfully higher than those of the IS. Clinical studies have demonstrated that using analgesic medication does not improve verbal and visuospatial memory in migraineurs [4,46]. Our published clinical observations also found that in the five (i.e., language, executive function, computational, memory, and orientation domains) domains of cognitive impairment in migraineurs, only executive function was associated with headache attacks [21]. We chose to assess cognitive function in mice with a novel object recognition test that primarily tests the animal’s memory capacity [19]. When subjects have executive dysfunction and intact memory capacity, the results of novel object recognition should show a decrease in total exploration time but a typical recognition ratio and recognition index. In our study, however, both indices were lower in migraine mice than in controls, and the differences were statistically significant, suggesting an inherent memory impairment in migraine mice. In addition, the results of the attention shift test also support this conclusion. It has been reported that there may be similar structural brain damage between adolescent migraine and attention deficit and hyperactivity disorders [47]. The migraine rats did not perform well in IDS, EDS, and REV stages, indicating that the formation, maintenance, and reversal of learning functions were impaired. After the sumatriptan treatment, only the performance of the IDS stage was recovered, and the attention shift of EDS and the reverse learning function of the REV stage did not change. This reminds us that the analgesic effect may have improved executive function but did not reverse the memory impairment. One possible explanation is that triptan analogs as acute analgesics have little effect on central migraine pathophysiological processes. As we know, triptan has both peripheral and central effects. In the periphery, triptans’ action at 5-HT1B receptors lessens pain brought on by intracranial vasodilation. In the central, triptans blocks the caudate nucleus of the trigeminal nerve’s afferent input of nociceptive signals centrally [48]. However, sumatriptan has difficulty crossing the blood–brain barrier to exert its central effects [49] and cannot block dural afferent inflammatory stimuli; this limitation means it can only relieve pain and restore pain-deprived attention, with little impact on central sensitization and cognitive impairment. Therefore, it can be speculated that memantine ameliorates cognitive impairment through an alternative pathway, and based on our testing of Arc, it appears to be the effector molecule through which memantine exerts this effect.

Since migraine causes increased and decreased synaptic plasticity in the trigeminal nerve and hippocampus, respectively, each lead to central sensitization and cognitive dysfunction [6,7,8]. Clinical cases of good results with memantine for migraine have been reported [50], and there are clinical studies to verify the efficacy of memantine as a prophylactic treatment for migraine [51,52]. It seems to be a promising direction to incorporate memantine into the treatment of migraine. However, compared to 5-HTR, CGRP, and PACAP, the consideration of NMDAR as a targeted treatment for migraine is still in its infancy [53], and migraine-related cognitive impairment is not mentioned in the latest expert consensus on migraine diagnosis and management [54].

## 5. Limitations

This study has some limitations that should be noted. There are so many sites associated with cognitive function that it is difficult to test them comprehensively. Changes in the prefrontal cortex and hippocampus alone are not sufficient to fully represent the pathophysiological processes involved in impaired cognitive function. In future studies, we will test site-specificity in more detail. Then, memantine, as an NMDA receptor antagonist, affects neurons by itself. This leads to difficulties in distinguishing between its synaptic restoring action mediated through Arc or its own activity. Furthermore, for reasons of avoiding estrogenic interference, only male mice were selected for this study, but females are more often affected by migraine, and further studies in our team are underway in clinical studies of female migraine patients and studies of estrogenic effects on migraine in female mice. Finally, the IS model mice were selected for cognitive function studies because previous studies have shown impaired recognition and spatial memory [55], but this model is only partially representative of migraine symptoms, and those symptoms analyzed with this model are only present in the chronic form, often from years of overuse of medications.

## 6. Conclusions

In this study, we observed changes in learning and memory functions, attention, hippocampal synaptic plasticity, and Arc protein expression in a mouse model of migraine induced by repeated administration of dural inflammatory stimuli. We conclude a possible pathophysiological process whereby migraine induces an upregulation of Arc expression in the hippocampus and cortex by some means—possibly the effect of inflammatory stimuli on glutamate receptors—which in turn causes a downregulation of presynaptic function and AMPAR-mediated synaptic scaling, leading to an external manifestation of homeostatic synaptic plasticity dysfunction and cognitive dysfunction. The application of memantine can restore these dysfunctions to some extent. However, we must also note that there are many symptoms migraineurs experience that remain outside of our research field, which means that there are still many pressing issues that need to be addressed.

## Figures and Tables

**Figure 1 brainsci-13-00331-f001:**
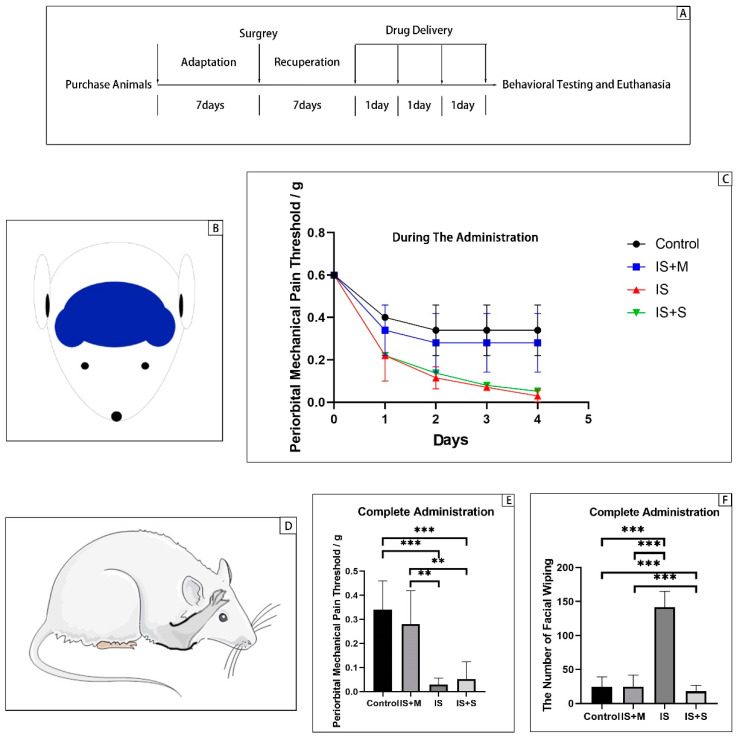
von-Frey mechanical pain threshold measurement and spontaneous pain behavior. Control: control group, n = 16; IS+M: IS + memantine group, n = 16; IS: inflammatory soup group, n = 16; IS+S: IS + sumatriptan group, n = 16. ***: *p* < 0.001, **: *p* < 0.01. (**A**): Experimental flow chart. (**B**): The blue area is the von-Frey mechanical pain threshold measurement area; (**C**): Changes in pain thresholds over time. Abscissa zero point is before surgery, and the day1 is the first day of administration. (**D**): Raised forepaws over the eyes and downward are considered to be spontaneous headache behaviors in mice. (**E**): The periorbital mechanical pain thresholds after administration was completed. (**F**): The facial wiping times after administration was completed.

**Figure 2 brainsci-13-00331-f002:**
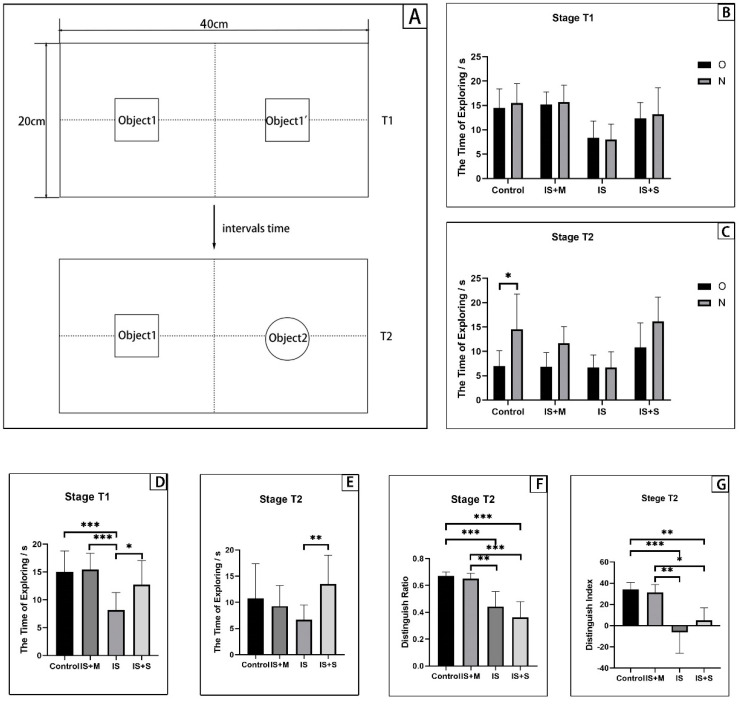
Novel Object Recognition Test. O: the time of exploring object 1; N: the time of exploring object 1’ or 2. ***: *p* < 0.001, **: *p* < 0.01, *: *p* < 0.05. (**A**): Schematic diagram of the experimental setup for novel object recognition. (**B**): The time spent exploring object 1 and object 1’ separately in stage T1. (**C**): The time spent exploring object 1 and object 2 separately in stage T2. (**D**): The sum of the time spent exploring object 1 and object 1’ in stage T1. (**E**): The sum of the time spent exploring object 1 and object 2 in stage T2. (**F**): Distinguish ratio at stage T2. (**G**): Distinguish index at stage T2.

**Figure 3 brainsci-13-00331-f003:**
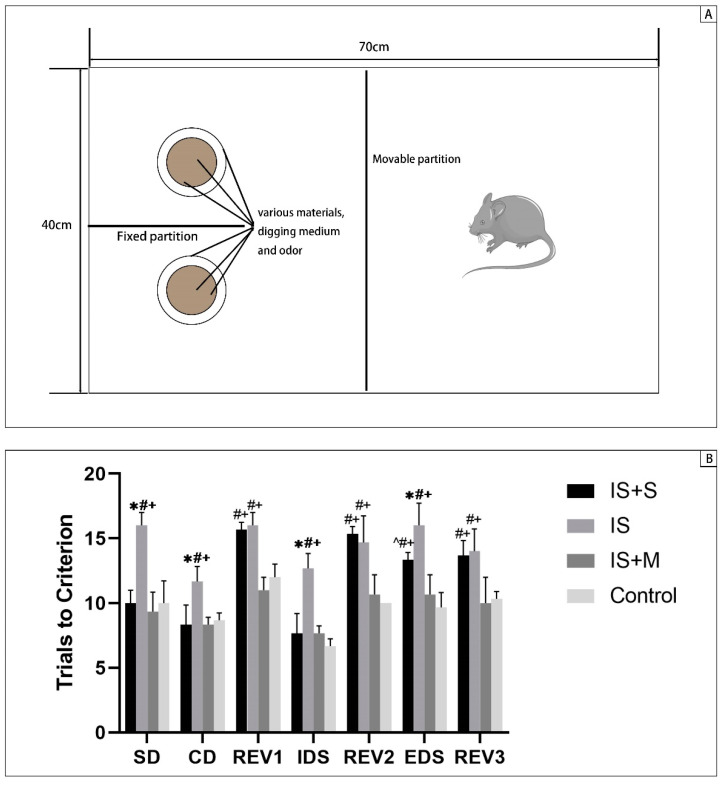
Attentional set-shifting test. *: *p* < 0.05, vs. the IS+S group; ^: *p* < 0.05, vs. the IS group; #: *p* < 0.05, vs. the IS+M group; +: *p* < 0.05, vs. the Control group. (**A**): Schematic diagram of the Attention Set Shifting experimental setup. (**B**): TTC at different stages in each group of mice.

**Figure 4 brainsci-13-00331-f004:**
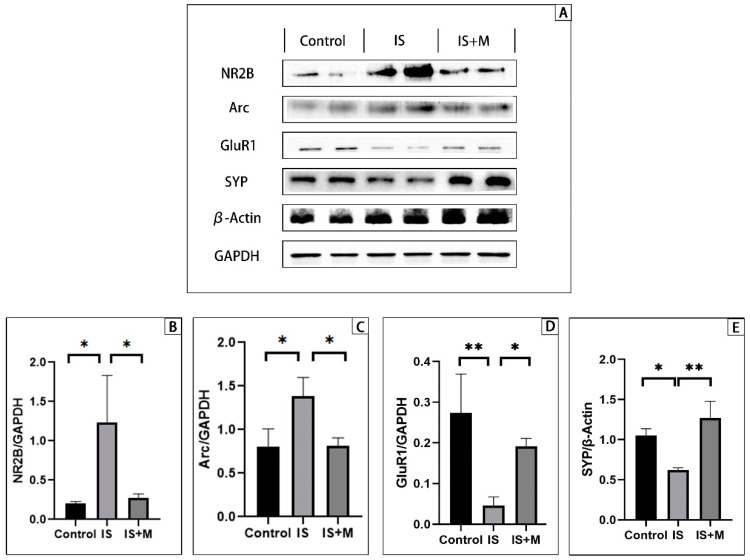
Determination of hippocampal protein in mice. **: *p* < 0.01, *: *p* < 0.05. (**A**): WB results of hippocampal protein. (**B**): NR2B/GAPDH ratio. (**C**): Arc/GAPDH ratio. (**D**): GluR1/GAPDH ratio. (**E**): SYP/β-Actin ratio.

**Figure 5 brainsci-13-00331-f005:**
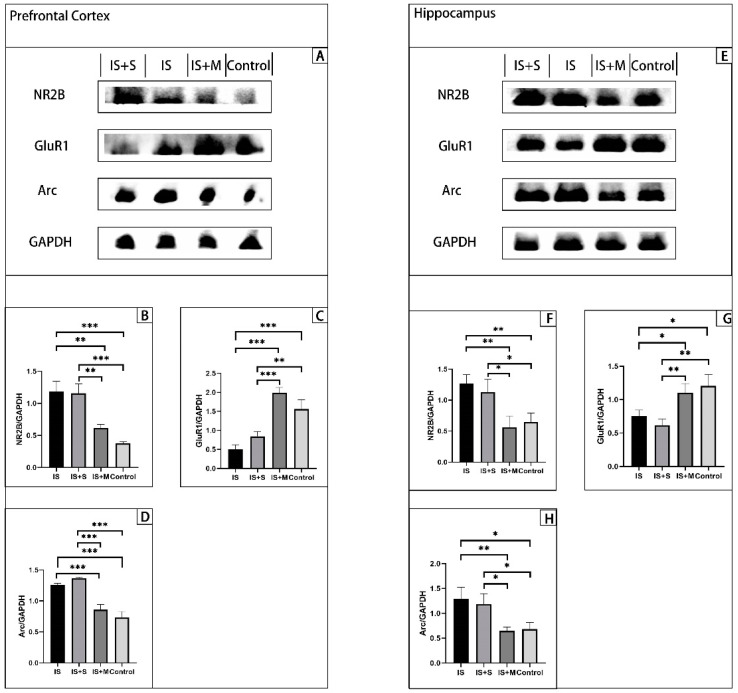
Determination of protein in the prefrontal cortex and hippocampus. ***: *p* < 0.001, **: *p* < 0.01, *: *p* < 0.05. (**A**): WB results of prefrontal cortex protein. (**B**): NR2B/GAPDH ratio. (**C**): GluR1/GAPDH ratio. (**D**): Arc/GAPDH ratio. (**E**): WB results of hippocampus protein. (**F**): NR2B/GAPDH ratio. (**G**): GluR1/GAPDH ratio. (**H**): Arc/GAPDH ratio.

**Figure 6 brainsci-13-00331-f006:**
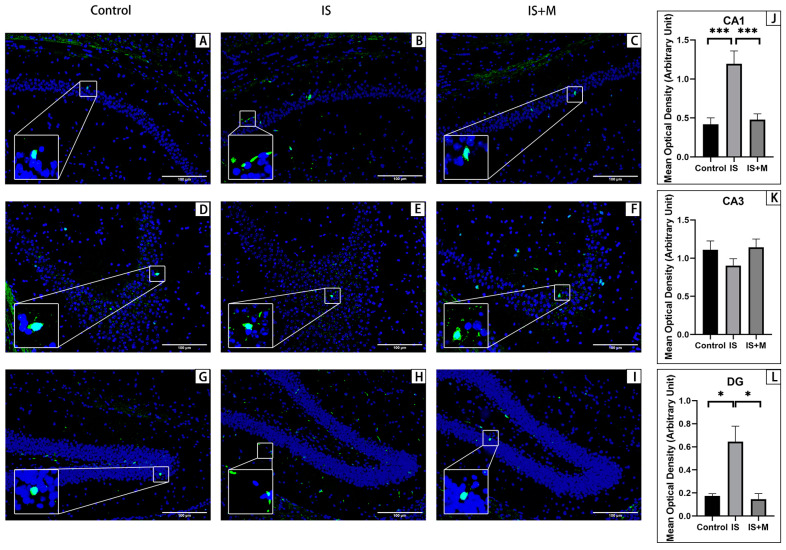
Immunofluorescence detection of Arc protein in hippocampus. ***: *p* < 0.001, *: *p* < 0.05. The blue light point in the figure is the nucleus. The green light dot is Arc, and the bottom left corner is a zoomed-in view of it. (**A**): Hippocampal CA1 region of the Control. (**B**): Hippocampal CA1 region of the IS. (**C**): Hippocampal CA1 region of the IS+M. (**D**): Hippocampal CA3 region of the Control. (**E**): Hippocampal CA3 region of the IS. (**F**): Hippocampal CA3 region of the IS+M. (**G**): Hippocampal DG region of the Control. (**H**): Hippocampal DG region of the IS. (**I**): Hippocampal DG region of the IS+M. (**J**): Comparison of mean optical density in the CA1 region. (**K**): Comparison of mean optical density in the CA3 region. (**L**): Comparison of mean optical density in the DG region.

**Figure 7 brainsci-13-00331-f007:**
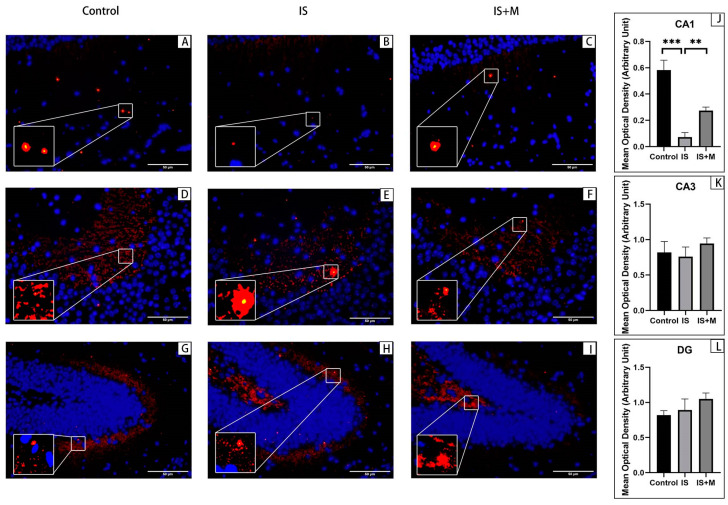
Immunofluorescence detection of SYP protein in the hippocampus. ***: *p* < 0.001, **: *p* < 0.01. The blue light point in the figure is the nucleus. The red light dot is SYP, and the bottom left corner is a zoomed-in view of it. (**A**): Hippocampal CA1 region of the Control. (**B**): Hippocampal CA1 region of the IS. (**C**): Hippocampal CA1 region of the IS+M. (**D**): Hippocampal CA3 region of the Control. (**E**): Hippocampal CA3 region of the IS. (**F**): Hippocampal CA3 region of the IS+M. (**G**): Hippocampal DG region of the Control. (**H**): Hippocampal DG region of the IS. (**I**): Hippocampal DG region of the IS+M. (**J**): Comparison of mean optical density in the CA1 region. (**K**): Comparison of mean optical density in the CA3 region. (**L**): Comparison of mean optical density in the DG region.

**Figure 8 brainsci-13-00331-f008:**
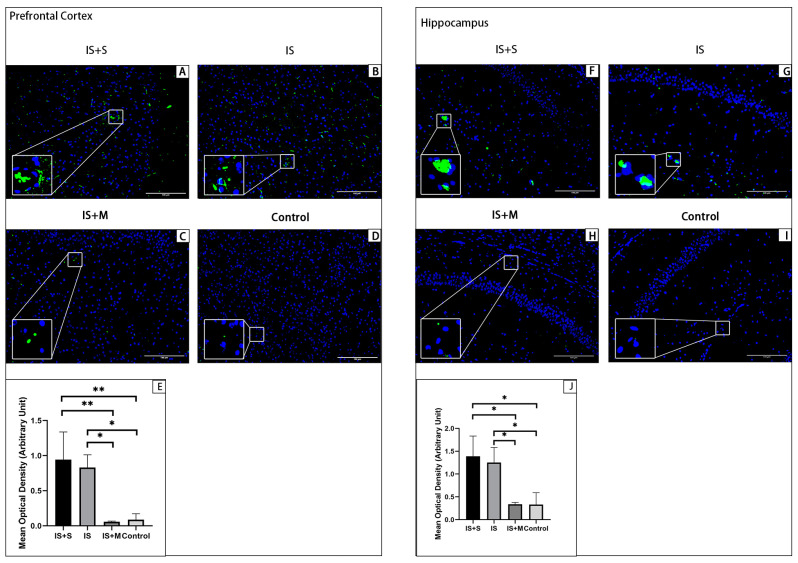
Immunofluorescence detection of Arc protein in the prefrontal cortex and hippocampal CA1 region. **: *p* < 0.01, *: *p* < 0.05. The blue light point in the figure is the nucleus. The green light dot is Arc, and the bottom left corner is a zoomed-in view of it. (**A**): Arc protein in the prefrontal cortex of the IS+S. (**B**): Arc protein in the prefrontal cortex of the IS. (**C**): Arc protein in the prefrontal cortex of the IS+M. (**D**): Arc protein in the prefrontal cortex of the Control. (**E**): Comparison of Arc’s mean optical density in each group’s prefrontal cortex. (**F**): Arc expression in the hippocampal CA1 region of the IS+S. (**G**): Arc expression in the hippocampal CA1 region of the IS. (**H**): Arc expression in the hippocampal CA1 region of the IS+M. (**I**): Arc expression in the hippocampal CA1 region of the Control. (**J**): Comparison of Arc’s mean optical density in each group’s hippocampal CA1 region.

**Figure 9 brainsci-13-00331-f009:**
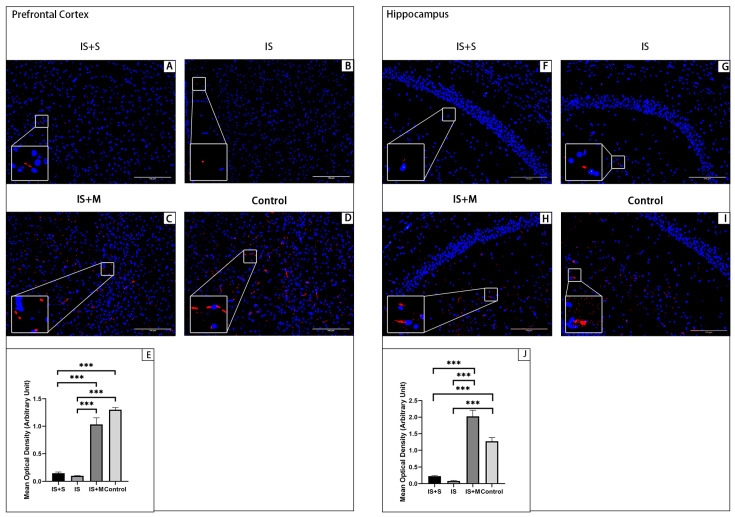
Immunofluorescence detection of GluR1 protein in the prefrontal cortex and hippocampal CA1 region. ***: *p* < 0.001. The blue light point in the figure is the nucleus. The red light dot is GluR1, and the bottom left corner is a zoomed-in view of it. (**A**): GluR1 protein in the prefrontal cortex of the IS+S. (**B**): GluR1 protein in the prefrontal cortex of the IS. (**C**): GluR1 protein in the prefrontal cortex of the IS+M. (**D**): GluR1 protein in the prefrontal cortex of the Control. (**E**): Comparison of GluR1’s mean optical density in each group’s prefrontal cortex. (**F**): GluR1 expression in the hippocampal CA1 region of the IS+S. (**G**): GluR1 expression in the hippocampal CA1 region of the IS. (**H**): GluR1 expression in the hippocampal CA1 region of the IS+M. (**I**): GluR1 expression in the hippocampal CA1 region of the Control. (**J**): Comparison of GluR1’s mean optical density in each group’s hippocampal CA1 region.

**Figure 10 brainsci-13-00331-f010:**
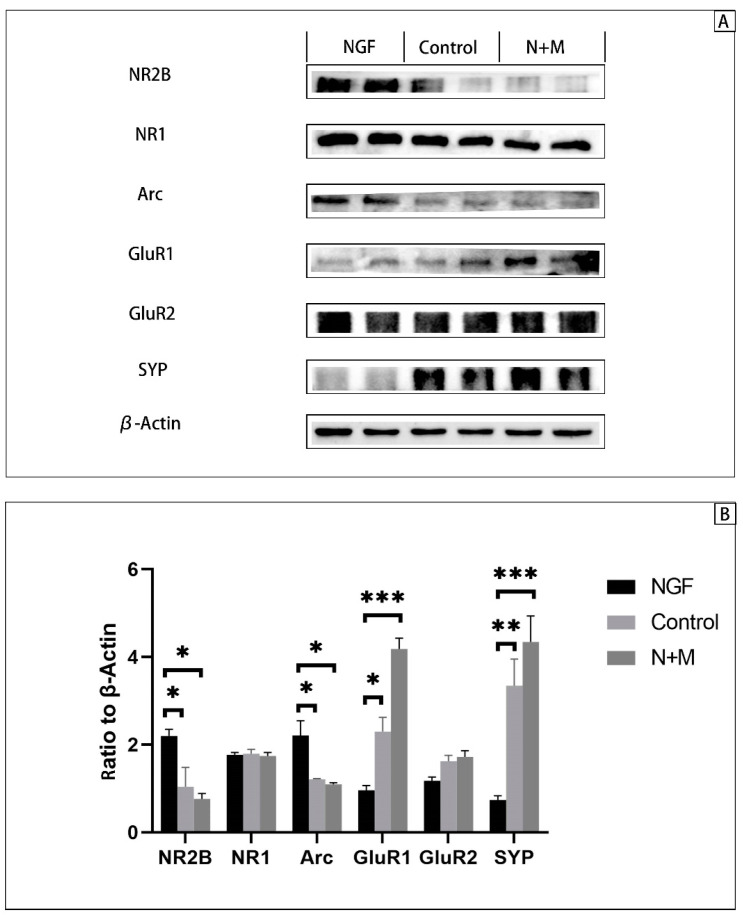
Cellular protein expression. NGF: nerve growth factor group, n = 4; Control: Control group, n = 4; N + M: NGF + memantine group, n = 4. ***: *p* < 0.001, **: *p* < 0.01, *: *p* < 0.05. (**A**): Western blot results of cultured cells. (**B**): The ratio of NR2B, NR1, Arc, GluR1, SYP to β-Actin, respectively.

**Figure 11 brainsci-13-00331-f011:**
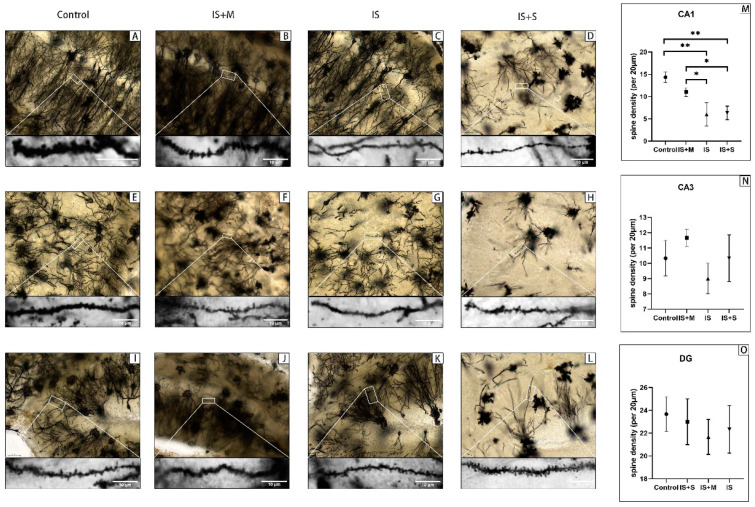
Golgi–Cox staining to detect the number of dendritic spines in hippocampal neurons in animal models. **: *p* < 0.01, *: *p* < 0.05. The black lines in the enlarged partial picture are dendrites, and the black dots on the dendrites are dendritic spines. (**A**): Hippocampal CA1 region of the Control. (**B**): Hippocampal CA1 region of the IS+M. (**C**): Hippocampal CA1 region of the IS. (**D**): Hippocampal CA1 region of the IS+S. (**E**): Hippocampal CA3 region of the Control. (**F**): Hippocampal CA3 region of the IS+M. (**G**): Hippocampal CA3 region of the IS. (**H**): Hippocampal CA3 region of the IS+S. (**I**): Hippocampal DG region of the Control. (**J**): Hippocampal DG region of the IS+M. (**K**): Hippocampal DG region of the IS. (**L**): Hippocampal DG region of the IS+S. (**M**): Comparison of spine density in the CA1 region. (**N**): Comparison of spine density in the CA3 region. (**O**): Comparison of spine density in the DG region.

**Figure 12 brainsci-13-00331-f012:**
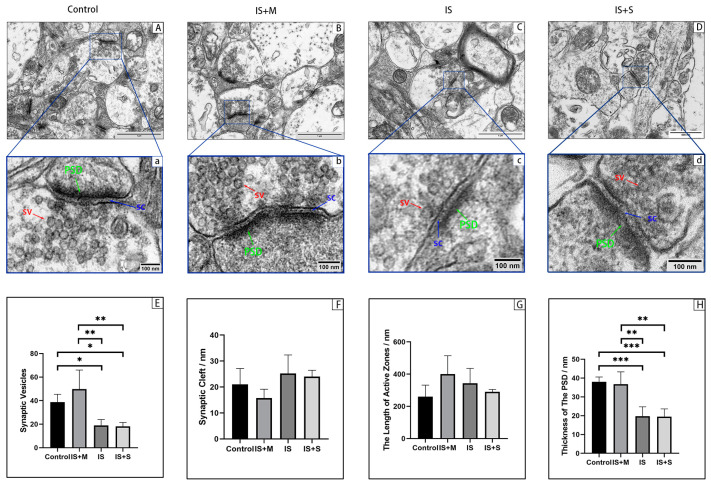
Synaptic Ultrastructure under Transmission Electron Microscope. ***: *p* < 0.001, **: *p* < 0.01, *: *p* < 0.05. SV: Red arrows show synaptic vesicles. SC: Blue arrows show the synaptic cleft. PSD: Green arrows show postsynaptic density. (**A**,**a**): Ultrastructure of hippocampal synapses in the Control. (**B**,**b**): Ultrastructure of hippocampal synapses in the IS+M. (**C**,**c**): Ultrastructure of hippocampal synapses in the IS. (**D**,**d**): Ultrastructure of hippocampal synapses in the IS+S. (**E**): Number of synaptic vesicles. (**F**): Width of synaptic cleft. (**G**): Length of active zones. (**H**): Thickness of PSD.

## Data Availability

All data generated or analyzed during this study are included in this article.

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
