# Peer review of "Arc-Mediated Synaptic Plasticity Regulates Cognitive Function in a Migraine Mouse Model"

_brainsci, 2023, doi:10.3390/brainsci13020331_

Round 1

Reviewer 1 Report

The study is very interesting, well structured and conducted. The data collected are many, methodologically correct and significant. The illustrations are beautiful and well explained.

There are some aspects that could be improved

Lines 39-40= The assumption is a bit exaggerated. Defining anxiety and depression and cognitive deficits as directly related to Arc-protein is still an interesting working hypothesis. Even defining these aspects as primary endpoints in migraine in this context is only a hypothesis of the authors.

Line 68= Why did you choose memantine over Sumatriptan? What is the relationship between these two compounds? While sumatriptan is specific only for the migraine attack, memantine is not used in the prevention of migraine or even the attack.

Line 127= please replace 'fibrils' with 'filaments'

Lines 689-90 = I agree with the authors who have clearly reported those specific to the study in the Limitations section. In particular it is necessary to underline the direct action of Memantine and the like on neurons. Secondly, the last sentence (line 689) of the section should be underlined as the accompanying symptoms of migraine are many and varied. Those analyzed with this model are present only in the chronic forms, often with overuse of drugs for many years.

lines 704-6= The last sentence of the Conclusions is risky and represents only a hypothesis of the authors and this aspect should be highlighted.

Author Response

Thank you for your comments, they have helped me a lot.

I have adjusted the description of the introductory section hypothesis to "it would be a good direction to take to improve cognitive function into account, rather than just focusing on pain, when developing new drugs for migraine."

Sumatriptan was chosen to test whether the current first-line drugs for migraine treatment help with cognitive symptoms, and memantine was chosen because we hypothesized that Arc has a key role in this cognitive impairment, and memantine can block any form of Arc overexpression [1]. There are a small number of reports of migraine treatment with memantine [2,3,4].

The term “fibrils” has been replaced with “filaments”.

The effect of memantine itself on neurons and the inability of the model to represent the complex clinical picture of migraine are highlighted in the limitations section.

The last sentence of the conclusion has been removed. Thank you for pointing this out.

[1] Steward O, Worley PF. Selective targeting of newly synthesized Arc mRNA to active synapses requires NMDA receptor activation. Neuron. 2001 Apr;30(1):227-40. doi: 10.1016/s0896-6273(01)00275-6. PMID: 11343657.

[2] Spengos K, Theleritis C, Paparrigopoulos T. Memantine and NMDA antagonism for chronic migraine: a potentially novel therapeutic approach? Headache. 2008 Feb;48(2):284-6. doi: 10.1111/j.1526-4610.2007.01016.x. Epub 2008 Jan 8. PMID: 18194287.

[3] Charles A, Flippen C, Romero Reyes M, Brennan KC. Memantine for prevention of migraine: a retrospective study of 60 cases. J Headache Pain. 2007 Sep;8(4):248-50. doi: 10.1007/s10194-007-0406-7. Epub 2007 Sep 24. PMID: 17901918; PMCID: PMC3451671.

[4] Bigal M, Rapoport A, Sheftell F, Tepper D, Tepper S. Memantine in the preventive treatment of refractory migraine. Headache. 2008 Oct;48(9):1337-42. doi: 10.1111/j.1526-4610.2008.01083.x. PMID: 19031499.

Reviewer 2 Report

The authors address from the point of view ofan animal experimentalist to bring light to the cognitive impairment present and demonstrated in the migraine sufferer. The design is sound, the methodology is tested and the results are interesting. Indeed, the translational results of this study alterations in Arc-mediated synaptic plasticity may be interpreted as an important mechanism of cognitive dysfunction in migraine.

I suggest just a few references for inclusion:

PMID: 35780081 

PMID: 35883043 

Author Response

Thank you very much for your suggestions. I have added these two articles to the citation." PMID: 35780081" citation number “[55]” and "PMID: 35883043" citation number “[20]”. Incidentally, the "PMID: 35780081" article is also the work of our team.

Reviewer 3 Report

The authors tried to investigate the pathophysiological mechanisms of cognitive dysfunction due to migraine in a migraine mouse mode. The topic is really interesting because is still poorly understood and not applied in clinical practice; moreover, the manuscript is clear and well written.

The Methods and Results sections are clear and full of details but I think that Discussion is the weak part of the manuscript: more work should be done in organizing this section; in particular, no explanation about the (possible) specific mechanisms is present. Moreover, no possible applications in the clinical evaluation of migraine patients is hypothesized: the authors should read and cite the paper by P Parisi et al. Epilepsy Behav. 2014;32:72-5. This aspects can be of interest for the readers.

Author Response

Thank you for your suggestion. We have refined the description of the possible mechanism of action of Arc in the discussion section and also mentioned case reports and clinical trials of memantine for migraine with cognitive impairment [2,3,4]. The paper by P Parisi et al. Epilepsy Behav. 2014;32:72-5 is cited with the citation number “[47]”.